# Two-Dimensional MFI Zeolite Nanosheets Exfoliated by Surfactant Assisted Solution Process

**DOI:** 10.3390/nano11092327

**Published:** 2021-09-07

**Authors:** Aafaq ur Rehman, Devipriyanka Arepalli, Syed Fakhar Alam, Min-Zy Kim, Jungkyu Choi, Churl Hee Cho

**Affiliations:** 1Reaction & Separation Nanomaterials Laboratory, Graduate School of Energy Science and Technology, Chungnam National University, 99 Daehak-ro, Yuseong-gu, Daejeon 34134, Korea; aafaqktk@cnu.ac.kr (A.u.R.); devipriya0508@o.cnu.ac.kr (D.A.); fakhar9689@o.cnu.ac.kr (S.F.A.); mzkim@cnu.ac.kr (M.-Z.K.); 2Department of Chemical and Biological Engineering, College of Engineering, Korea University, 145, Anam-ro, Seongbuk-gu, Seoul 02841, Korea; jungkyu_choi@korea.ac.kr

**Keywords:** MFI zeolite, two-dimensional zeolite, exfoliation, PVP, colloidal stability

## Abstract

Two-dimensional (2D) zeolite nanosheets are important for the synthesis of high flux zeolite membranes due to their lateral size in a preferred orientation. A way to obtain 2D zeolite nanosheets is to exfoliate interlocked structures generated during the hydrothermal synthesis. The mechanical and polymer assisted exfoliation process leads to mechanical damage in nanosheets and short lateral size. In the present study, polyvinylpyrrolidone (PVP) was introduced as an exfoliation agent and dispersant, so that multilamellar interlocked silicalite-1 zeolite nanosheets successfully exfoliated into a large lateral size (individual nanosheets 500~1200 nm). The good exfoliation behavior was due to the strong penetration of PVP into multilamellar nanosheets. Sonication assisted by mild milling helps PVP molecules to penetrate through the lamellar structure, contributing to the expansion of the distance between adjacent layers and thus decreasing the interactions between each layer. In addition, the stability of exfoliated nanosheets was evaluated with a series of organic solvents. The exfoliated nanosheets were well dispersed in *n*-butanol and stable for 30 days. Therefore, the PVP-assisted solution-based exfoliation process provides high aspect ratio MFI zeolite nanosheets in organic solvents for a long period.

## 1. Introduction

Zeolites are silica-based microporous materials with pore size range between 0.25 to 2 nm. For optimal catalysis and separation, it is crucial to control the pore network to allow the targeted molecular traffic in the pores [1]. However, challenges ahead of controlling the preferred orientation of the zeolite particles remain a significant concern among zeolite researchers. Furthermore, the ultra-thin zeolite films and nano-size zeolite sheets help in various processes, such as high-flux membrane separations, catalysis, sensors and the adsorption process [2,3]. Most of these challenges are overcome by decreasing their thickness to a few tens nm, which are called two-dimensional (2D) zeolites. Two-dimensional zeolites are very promising materials for catalysis due to reduction in diffusion length, enhancement in the catalytic activity and increase in the lifetime of zeolite catalysts [4]. Two-dimensional zeolites offer attractive structural features in a wide range of other applications due to their ability to form thin coatings, short diffusion pathways for the selective separations of chemical species and high aspect ratio for the oriented growth of zeolite membranes [5].

There are two kinds of 2D zeolite nanosheet preparation methods: one is the bottom-up growth process, and the other is top-down. Commonly known as direct-synthesis, the bottom-up process of zeolite nanosheets involves nucleation and epitaxial growth into large lateral size nanosheets [6]. Sometimes, seed crystals are introduced to help the formation of 2D zeolites. However, the center of directly synthesized nanosheets are composed of seeds, which deteriorate the homogeneous thickness of membranes during the membrane preparation and thus create the possibility for non-zeolitic pores and defects in the membranes. The top-down process involves anisotropic etching of three-dimensional (3D) zeolite particles, disintegration of nanosheets and exfoliation into a few layers [7]. In recent years, research has gained attention in the synthesis of thin zeolite membranes through the exfoliation of layered and pillared structures, such as nanosheets or direct growth of nanosheets on the substrate [6,8]. Exfoliation of zeolite nanosheets is vital for fabricating thin zeolite films and reducing the thickness ten-fold to nanometers. Another important aspect of exfoliating the nanosheets is higher selectivity and permeance by reducing the inter-crystalline gap between the surfaces and controlling the crystallographic orientation in zeolite membranes [2,3,9]. Furthermore, the high performance of thin membranes is because of facilitating the molecular diffusion in straight channels along with the smaller thickness, which is helpful in the selective separation of smaller species [10,11].

ZSM-5 is a type of MFI zeolites that has a pentasil structure, patented by Mobil researchers in the 1970s. Since then, ZSM-5 has been widely used as a catalyst and adsorbent in fine chemical industries and membrane separation processes. Two-dimensional MFI zeolites are typically 2 nm thick in *b*-dimension and 100–1500 nm in *a-* and *c*-dimensions [12] and synthesized in pillared morphologies and thin films [5,13]. In the progress of the exfoliation of zeolites, Ogina et al. [14] exfoliated MCM-22 to create new material UCB-1 by delamination using an aqueous solution of cetyl trimethyl ammonium bromide (CTAB), TBAF and TBACl in ultrasonication. Varoon et al. [15] successfully exfoliated multilamellar MFI zeolite nanosheets by melt compounding method in the presence of active cationic polymer, such as polystyrene. The exfoliated nanosheets obtained with this method had a high aspect ratio and showed excellent performance for the fabrication of MFI membrane toward the selective separation of *p*-xylene/*o*-xylene. The MFI zeolite membrane obtained from exfoliated nanosheets showed a separation factor of 70. However, non-exfoliated particles existing in the suspension were still observed and have been reported for organic solvents such as *n*-octanol, cyclohexane, *n*-hexane, chloroform, dichloromethane, chlorobenzene, ethanol and *n*-butanol. Among organic solvents, *n*-octanol dispersed the nanosheets effectively and the resultant solution was stable for many days [16]. Maheswari et al. [17] successfully exfoliated MWW zeolite through the swelling with the help of CTAB under mild temperature and lower basic conditions. The swelling of layered materials to obtain exfoliated nanosheets is favorable in MWW, ITQ-2 and MCM-22 zeolites. However, MFI with a layered structure is hardly swelled by cationic surfactants, such as CTAB or tetrapropylammonium hydroxide (TPAOH), because it is already swelled by SDA anchored in the inside or associated with the zeolite layer [1]. Recently, Sabnis et al. [18] exfoliated the layered MCM-22 and multilamellar MFI nanosheets by suspending in telechelic liquid polybutadienes followed by sonication and shearing at room temperature. Their results indicate that the end groups of polybutadienes play a crucial role in the exfoliation process. Through the liquid processable technique, MCM-22 was successfully exfoliated. However, multilamellar silicalite-1 nanosheets were not fully exfoliated. Liquid phase exfoliation of layered materials is preferred over other techniques (melt compounding and swelling) because of the scalable, versatile and sustainable route for large-scale production [19,20]. An ideal exfoliated nanosheet should be free from embedded polymer, with minimum wrinkling, be easily dispersible and maintain stable suspension for many days. Furthermore, it creates the opportunity to be easily coated on supports with suspended nanosheets, while other techniques are almost difficult or impossible. From the literature survey, it has been concluded that the exfoliation of multilamellar silicalite-1 nanosheets (MLSil-1Ns) to large lateral size individual nanosheets and stable suspension of nanosheets are still challenging.

Recently, a large number of scientific reports have described the stability and exfoliation property of polyvinylpyrrolidone (PVP) as a nonionic surfactant. The exfoliation capability of PVP has gained a lot of interest and has been used as a surfactant or exfoliating agent in the exfoliation of phosphorene from black phosphorous [21], in graphite for the production of graphene [22], molybdenum disulfide nanosheets from natural molybdenite (MoS_2_) [23], boron nitride nanosheets (BNNS) and tungsten disulfide (WS_2_) [24]. Furthermore, PVP has been widely used as a stabilizing and penetrating agent. It provides excellent load transfer and retains the mechanical stability of layered materials [25]. Therefore, the main objective of this study is to exfoliate MLSil-1Ns in a larger lateral size using the promising nonionic surfactant, PVP.

In this study, we prepared crystalline MLSil-1Ns using diammonium structure-directing agent and investigated morphology evolution during the hydrothermal preparation. The prepared MLSil-1Ns were exfoliated with the help of PVP surfactant into large lateral size nanosheets. We studied the effect of PVP on the exfoliation of MFI nanosheets, and finally, we present the stability of exfoliated nanosheets in organic solvents.

## 2. Experimental

The bi-functional organic surfactant directing agent (OSDA), di-quaternary ammonium type C_22_H_45_-N^+^ (CH_3_)_2_-C_6_H_12_-N^+^(CH_3_)_2_-C_6_H_13_ (denoted as C_22-6-6_)_-_, was synthesized by the method developed by Ryoo and co-workers [9]. The detailed information about chemicals and experimental is provided in Appendix A. Briefly, before exfoliation, the synthesized MLSil-1Ns dried product was calcined at 550 °C for 5 h to remove the OSDA located in the crystalline materials. The exfoliation of MLSil-1Ns was conducted by mild milling in the presence of PVP as a surfactant. PVP is highly stable in aqueous solution [26]. In typical exfoliation, 10 wt.% stable suspension of PVP was separately prepared in D.I. H_2_O. In next step, zeolite nanosheets were added in the surfactant suspension in the ratio of 0.16 g of zeolite nanosheets and 3.84 g to 10 wt.% surfactant suspension and sonicated for 1 h using sonicator (Powersonic 605, 350 watts). The sonicated samples were subjected to rapid exfoliation under mild planetary milling of 400 rpm for 15 min (FRITSCH Ball mixer/mill15).

In a typical cleaning process, the exfoliated solution with PVP was dispersed in toluene to 1~5 wt.% suspension by sonication for 1 h. The resulting solution was centrifuged in two 50 mL centrifuge tubes at 18,000 rpm for 40 min to sediment the MLSil-1Ns at the bottom of the centrifuge tubes. Sedimented MLSil-1Ns were slowly and carefully separated from the supernatant and re-dispersed in toluene. This process was repeated three times to completely remove the injected PVP in the nanosheets. Finally, the sedimented MLSil-1Ns were obtained by pouring out the supernatant and bottom products were dispersed in organic solvating agents, such as *n*-butanol, chloroform and *ortho*-dichlorobenzene (*o*-DCB) and *N*′*N*′ dimethylformamide (*N*′*N*′DMF), to observe the suspension quality. Furthermore, to check the dispersible exfoliated nanosheets in the suspension for easy fabrication of thin films, characterization and quantification of nanosheets were performed to check the effect of various parameters.

Powder X-ray diffraction (XRD) patterns were obtained using PAN analytical X-pert Pro diffractometer with Cu Kα radiation. Nitrogen adsorption and desorption isotherm were measured using BELSORP-max Japan Inc., at the liquid N_2_ temperature of 77.35 K. Scanning electron microscope (SEM) images were recorded after Pt coating by using cold type SEM, Hitachi S-4800, Japan. Transmission electron microscope (TEM) study of nanosheets was carried out by FEI Tecnai G^2^ F30 S-Twin operated at 300 kV.

## 3. Results and Discussion

### 3.1. Effect of OSDA on the Formation of MLSil-1Ns

The as-synthesized C_22-6-6_ was used as OSDA in the synthesis of MLSil-1Ns. To investigate the effect of the synthesis time on crystallization, samples were hydrothermally synthesized for 5, 10 and 15 d at 150 °C. The XRD patterns are shown in Figure 1. The 5-d sample was amorphous, and no diffraction peaks were detected. With further increasing of the crystallization time to 10 d, very small XRD diffraction peaks of MFI crystal started appearing at 2θ of 23.5~24.5° and 30° but some peaks of MFI zeolite were not observed. The intensity of peaks was small so that the sample was mostly composed of disordered amorphous phases. Finally, sharp MFI diffraction peaks were observed from the 15-d sample. The peaks were broader as compared to standard MFI diffraction peaks for powder ZSM-5. All XRD peaks show the characteristics of the MFI zeolite framework structure, such as (101), (200), (301), (501) reflections which are associated with crystallographic *a-c* plane dimensions. The preferred crystallographic plane dimension confirms the absence of the *b*-dimension in the synthesized nanosheets. The crystal growth along the *b*-dimension is forbidden because of the presence of bulk tails of the surfactant molecules covering the *a-c* planes, which were efficiently assembled in a multilamellar array through hydrophobic interactions between surfactant tails outside the zeolite layers [27]. Similar results have been reported for multilamellar MFI nanosheets featuring the broadening of peaks compared to that of conventional MFI crystals [7,26,27].

The SEM images shown in Figure 2 provide detailed insights into the morphology of silicalite-1 nanosheets which were in multilamellar stacking with the three-dimensional self-interlocked structure. The particles obtained at 150 °C for 5 d were agglomerated irregularly and no clear morphology was obtained. Generally, a long hydrothermal time of more than 10 d is reported for complete multilamellar silicalite-1 nanosheets synthesis in the absence of organic co-solvents. The reason for long transformation to nanosheets is because of the di-quaternary ammonium head group in OSDA which strongly attracts silicate species [8]. The SEM images reveal the slow and insufficient growth of nanosheets due to a long chain attached with organic surfactant which hinders the nucleation of crystal to create the lamellar. Further extending the hydrothermal synthesis to 10 d, thick plate-like growth was observed. The thickness of each sheet and plates were large in size (700–900 nm), as shown in Figure 2c,d. These intermediate large plates will transform into nanosheets. According to the XRD pattern, the phase transformation degree to nanosheets is still very low. Finally, the samples synthesized at 150 °C for 15 d show complete nanosheet-like morphology composed of intergrown three-dimensional nanosheets in the interlocked structure, as shown in Figure 2e,f. The longer time to obtain lamellar stacking of layered zeolites suggests that the layer formation is retarded at the early stages of synthesis [28]. Therefore, to achieve completely intergrown three-dimensional zeolite nanosheets, the hydrothermal time might be continued to more than 10 d which could lead to the transformation from disordered unilamellar structure to multilamellar zeolite [27]. Even though many reports announced that hierarchical MFI zeolite sheets were synthesized at high concentrations of alkaline sources and organic co-solvents [27,28,29], hierarchical nanosheets were obtained without alkaline source and organic co-solvents in the present study. In the zeolite community, the study on the alkaline sources and organic co-solvents are well understood and discussed in numerous research articles [29,30].

Phase transformation through rotational intergrowth plays an important role because, in the presence of complex OSDA, morphology transformation usually occurs [31]. During the early stages of crystallization, the transformation phase is dominated by the amorphous phase, as evident in our study. The amorphous phase is comprised of thick plates in aggregate form. Even though the formation of plates is still unclear, the plate morphology announced to us that a considerable small number of zeolites are there. Perhaps the small zeolites can be attributed to the formation of plates. Interestingly, thin nanosheets were obtained from the plates with further increasing the time to 15 days [32].

The N_2_ adsorption-desorption isotherms and pore size distribution of MLSil-1Ns-15d are presented in Figure 3a. The adsorption and desorption isotherm obtained from calcined sample MLSil-1Ns-15d illustrates type IV adsorption isotherm. The abrupt adsorption at ultralow pressure and the hysteresis loop at relative pressure of 0.5 confirm the micro and mesoporous structures of the calcined sample [33]. It exhibited a reasonably high Brunauer–Emmett–Teller (BET) surface area of 309.25 m^2^ g^−1^. Before calcination, the pore volume of MLSil-1Ns was observed to be 0.30 cm^3^ g^−1^. After the calcination, the pore volume increased to 0.38 cm^3^ g^−1^. The adsorption isotherm of the sample before calcination was observed as the same type with low N_2_ adsorption, indicating the OSDA was present on the surface of the nanosheets but not in the framework. The BET surface area of the sample before calcination was observed to be 238.10 m^2^ g^−1^. The presence of mesoporosity in the lamellar silicalite-1 nanosheets are due to the large number of intergrown lamellae acting as a “pillar” to support each other, therefore preventing the collapse of the mesoporous structure [9].

The hierarchical structure of MLSil-1Ns could be confirmed with the pore size distribution. The microporous sharp peak was observed at a pore diameter of below 1 nm and mesoporous peak at 150 nm in the calcined sample, while the sample before calcination showed micropore peaks at 1.024 nm and mesopore peaks were not observed, as shown in Figure 3b. This indicates that OSDA present in the MLSil-1Ns restricts the adsorption of molecular species. Especially, OSDA present in the before-calcination sample retards capillary condensation in the mesopores but does not affect adsorption in the micropores [34]. The results announce that OSDA presents in the interparticle voids between the MLSil-1Ns or interlamellar voids, not in the micropores. On considering the size and preferential positioning of OSDA on the *a-c* plane, it is expected that OSDA presents on the surface of nanosheets and in mesopores. In all exfoliation experiments, a calcined sample without OSDA was used.

### 3.2. Effect of PVP on the Exfoliation of MLSil-1Ns

The delaminated structure of MLSil-1Ns increases the accessibility of small molecules to the micropores due to the shortened diffusion pathways by reduced thickness and the growth of membranes in a preferred orientation (*b*-axis in the case of MFI membrane). To attain the proper mixing of zeolite nanosheets and surfactant (PVP), the mixture was sonicated for 1 h. Secondly, the sonicated mixture was exposed to planetary milling for 15 min at room temperature. For comparison, the as-synthesized unexfoliated sample is shown in Figure 4a. The PVP embedded sample was obtained after exposure to 15 min milling with clearly no sign of MLSil-1Ns characteristics, which means the surfactant has been completely distributed in the nanosheets (Figure 4b). The obtained products were extensively purified by the dissolution process using toluene to remove embedded surfactants. The Sil-1Ns with much higher density settled down at the bottom of the toluene suspension. Due to its readily soluble properties of PVP in water and organic solvent, [14] the introduced PVP into MLSil-1Ns dissolves in toluene, therefore the supernatant is separated from the sedimented nanosheets. However, with one-time dissolution, the PVP still exists in the nanosheets and therefore the nanosheets were agglomerated together, as shown in the SEM image (Figure 4c).

To further purify the nanosheets, dissolution was repeated two more times to further dissolve surfactant and then the nanosheets were separated using centrifugation. The nanosheets had a few or single sheet-like characters. However, a small fraction of surfactant was observed after two-time centrifugations, as shown in SEM image Figure 4d. During the centrifugation of exfoliated nanosheets, the surfactant embedded in the nanosheets matrix was removed, leaving at the top of the centrifuge tube. This process of multiple dissolutions reduces the concentration of surfactant by dissolving in toluene, and purified Sil-1Ns could be obtained, as shown in Figure 4e,f. The dissolution process purifies the Sil-1Ns free from surfactant. PVP molecules can physiosorbed into the surface and inter-surfaces of multilayered zeolites [24]. The important characteristic of surfactant-assisted exfoliated nanosheets in the present study is the large lateral size of each nanosheet. Figure 4e,f shows the large lateral size of nanosheets exfoliated from the interlocked structure typically in the range of 500~1200 nm. The surfactant physiosorbed between the multilamellar materials enhanced the exfoliation of nanosheets [35]. Since both Sil-1Ns and PVP are strongly hydrophobic, they are considered one unit [22,36,37]. The hydrophilic amide group of PVP points toward the continuous aqueous phase, whereas the hydrophobic methylene chains direct towards the nanosheets side. We hypothesized that silicalite-1 nanosheets might exist on the hydrophobic side [38,39]. With further repetition of centrifugation, it was easy to observe the nanosheets. This could be attributed to the removal of PVP surfactant from the solution by leaving behind the exfoliated nanosheets in a few or single layers.

Liquid phase exfoliation assisted by surfactant behaves in two different steps. In the first step, the weaker interlayer attractions between adjacent nanosheets are overcome by the chemical energy released during the PVP penetration. In the second step, the nanosheets are stabilized by surfactant to avoid re-aggregation. The surfactants introduced into the inter-surfaces decreased the energy required for the exfoliation and prevented them from re-stacking. The stabilization of nanosheets by surfactant is important because they minimize the net energy required for the exfoliation by adsorbing on the surface of nanosheets and shielding them from re-stacking in a liquid medium. Another important function of PVP is to sustain the high aspect ratio during the exfoliation. The exfoliated nanosheets are usually diverse in lateral size and thicknesses [40]. Therefore, PVP aid in the weaker interlayer attraction between adjacent nanosheets.

One important feature of PVP as nonionic surfactant is its ability to penetrate layered materials through strong hydrophobic interactions between layered materials and PVP chains [41]. The stabilization results in decreasing the rate of re-stacking [40]. The exfoliation of layered materials occurs whenever the bond between layered materials is overcome by the intercalants. In this regard, PVP surfactant could be a good candidate for the exfoliation of 2D materials due to its low cost, relatively high output and yields of large aspect ratio of nanosheets.

### 3.3. Effect of Solvating Agents on Dispersion of Sil-1Ns

After substantial repetition of dissolution with toluene, the exfoliated Sil-1Ns were suspended in organic solvents as shown in Figure 5. To make a stable suspension, we explored organic solvents such as chloroform, *n*-butanol, *N*′ *N*′ DMF and *o*-DCB. SEM images show the exfoliated nanosheets clearly. The exfoliated nanosheets have a lateral size in the range of 500–1500 nm. The dispersion of these nanosheets varied with solvents. Exfoliated nanosheets dispersed in *N*′ *N*′ DMF and chloroform settle down after a few hours and sediment at the bottom of the suspension, as can be seen in the dispersion solution images. On the other hand, nanosheets were well dispersed in *n*-butanol and *o*-DCB for a long time. Figure 6 shows the TEM images of exfoliated nanosheets dispersed in organic solvents. As observed in TEM images, exfoliated nanosheets were freely suspended in the *n*-butanol and *o*-DCB as shown in Figure 6a,b and Figure 6d,e, respectively. The corresponding selected area electron diffraction (SAED) patterns confirm the high crystallinity and presence of exfoliated nanosheets because the nanosheets are lying flat on the surface of the TEM grid. The presence of diffraction spots in the *n*-butanol sample (Figure 6c) confirms that the exfoliated nanosheets are highly crystalline along the *b*-orientation due to small *d*-spacing as 2.0 Å [16]. Almost similar SAED patterns were observed in *o*-DCB solvent with high crystallinity and small d-spacing, as shown in Figure 6f. In contrast, slightly agglomerated nanosheets were observed in the *N*′*N*′DMF solvents. The corresponding ED patterns were much sharp and denser as compared to *n*-butanol (Figure 6g–i). Interestingly, chloroform suspended Sil-1Ns shows the re-stacking of nanosheets and corresponding ED patterns confirm the agglomeration due to the overlapping of nanosheets (Figure 6j–l). It can be observed from TEM images that *n*-butanol and *o*-DCB not only maintain stability but also crystallinity of nanosheets, while little to no stability of Sil-1Ns was observed in *N*′*N*′DMF and chloroform solvents (Figure 6i,l).

The purified and exfoliated nanosheets dispersed in *n*-butanol, suspended for a longer time and remained separated from each other. This phenomenon of stable suspension for a longer time of nanosheets can be explained by classical DLVO theory and steric hindrance. Without concern to the origin of repulsion forces, there is always the van der Waals attraction. The repulsion can be achieved by increasing the charge on the nanoparticle or by adding larger molecular side chains to induce steric hindrance. Moreover, two repulsive forces can be combined [42]. In the present study, steric hindrance seems an obvious reason because *o*-DCB and *n*-butanol provide higher steric hindrance between the solvent and the zeolite nanosheets, therefore hindering the nanosheets from agglomeration. In the case of *o*-DCB, the steric hindrance is generated from a strong electron cloud in the benzene ring and has a higher steric value of 3.0 kCal mol^−1^ (Table 1). While in the case of *n*-butanol (1.70 kCal mol^−1^), the hindrance comes from the extended alkyl-chain, results in a strong barrier and prevents the nanosheets from agglomeration. Contrarily to *o*-DCB and *n*-butanol, chloroform and *N*′*N*′DMF exhibit smaller molecular weight as well as lower steric hindrance. Therefore, the agglomerated nanosheets settled down in these solvents.

In Table 2, the results of the present work are compared with previous reports for the exfoliation of zeolite nanosheets by melt compounding, swelling, milling and sonication. MCM-22 and ITQ-2 can be easily transformed into nanosheets simply by the swelling in the presence of strong cationic surfactant. However, the melt compounding method for the exfoliation of MFI nanosheets results in the crumbling and deterioration of nanosheets due to uncontrolled mechanical force from an extruder at high temperature (150 °C). Similarly, in the case of hydroxyl-terminated polybutadiene (HTPB), their results indicate that MFI nanosheets are difficult to exfoliate due to the sticky nature of HTPB. Based on the experimental results, a possible mechanism is proposed for the exfoliation of MLSil-1Ns in the PVP system, as shown in Figure 7. The di-ammonium group acts as structure directing for the formation of MFI crystal while the long-chain alkyl group creates the lamellar assemblies [43]. The hydrophilic amide groups of PVP surfactants point towards the interlamellar structure of interlocked nanosheets. During sonication, the hydrophobic chains dissolve inside the interlocking structure of nanosheets and PVP penetrates inside the MLSil-1Ns. During sonication, the PVP molecule also penetrates through the lamellar structure contributing to the expansion of the distance between adjacent layers and thus decreasing the interactions between each layer. It is noteworthy that PVP molecules play a crucial role by acting as a dispersing agent and a separating agent to prevent the aggregation of exfoliated Sil-1Ns. Optimal concentration of PVP is required to obtain curl-free nanosheets [23]. Furthermore, the electrostatic repulsive forces generated during the penetration of surfactants result in the MLSil-1Ns to swell and separate individually by mild milling. Meanwhile, single or few layers of Sil-1Ns are exfoliated from the surface of multilamellar nanosheets by the penetration of surfactant. The surfactant adsorbs onto the surface of exfoliated MFI nanosheets, providing steric hindrance against restacking and structural deterioration during milling. After exfoliation is achieved, repeated centrifugation and dissolution in toluene resulted in pure and washed nanosheets, free from cationic surfactant. Finally, the nanosheets remain stable for a long time in *n*-butanol due to strong steric hindrance behavior against zeolite nanosheets.

## 4. Conclusions

In conclusion, attempts to exfoliate the interlocked structure of MFI zeolites have been made to obtain high aspect ratio nanosheets without disturbing the crystallinity. The obtained nanosheets with a large lateral aspect ratio of MFI nanosheets were exfoliated with the help of PVP surfactant embedded in the nanosheets by mild milling. Sonication assisted by mild milling helps PVP molecules to penetrate through the lamellar structure, contributing to the expansion of the distance between adjacent layers and thus decreasing the interactions between each layer. The surfactant penetrates through the interlocked structure of multilamellar MFI nanosheets and exfoliates the nanosheets in a few or single layers. The TEM images demonstrate the crystallinity and sharp diffraction patterns of exfoliated nanosheets. The removal of surfactant was performed with the help of dissolution in toluene by repeated centrifugation. Finally, a stable suspension was obtained by exploring various organic solvents. Strong steric hindrance generated from the alkyl chain of *n*-butanol improves the stability of nanosheets for a longer period. This work provides new understanding of PVP-assisted liquid exfoliation and dispersion of 2D MFI zeolite nanosheets in various organic solvents.

## Figures and Tables

**Figure 1 nanomaterials-11-02327-f001:**
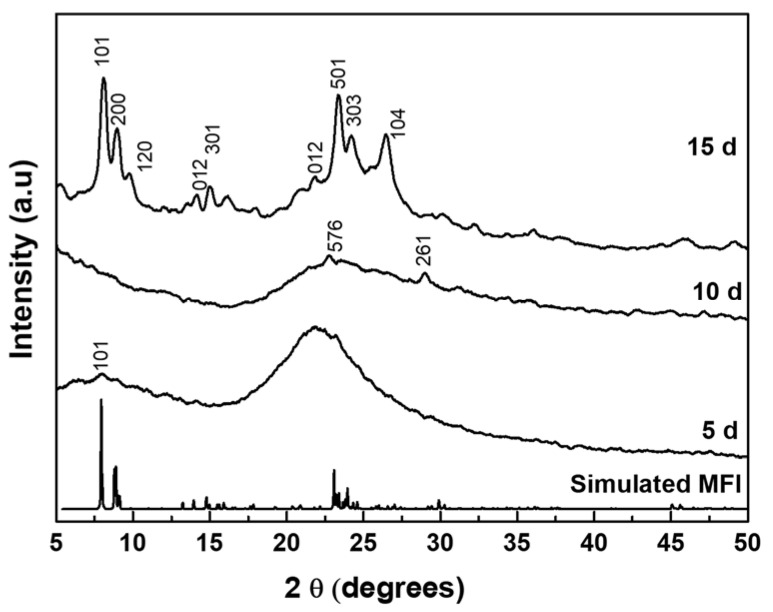
XRD diffraction patterns of powder samples synthesized at various hydrothermal times.

**Figure 2 nanomaterials-11-02327-f002:**
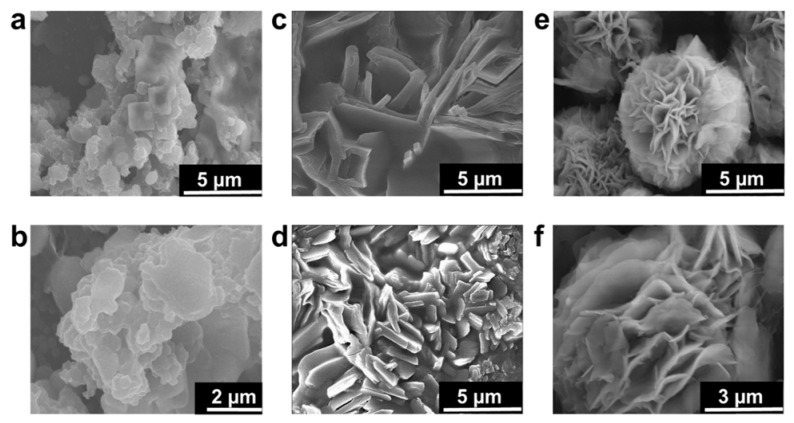
As-synthesized samples of MLSil-1Ns-x (where “x” is the number of days) (**a**–**f**). (**a**,**b**) SEM images of MLSil-1Ns-5d synthesized at 150 °C for 5 days. SEM images show that there is no growth of nanosheets. The particles are mostly agglomerated in irregular and spherical order. (**c**,**d**) SEM images of MLSil-1Ns-10d synthesized at 150 °C for 10 days. SEM images show that plate-like particles were formed, the thickness of each sheet was higher in size and intergrowth was not properly attained. (**e**,**f**), SEM images of MLSil-1Ns-15d synthesized at 150 °C for 15 days. SEM images show the MLSil-1Ns are plates and sheets like morphology in three-dimensional interlocked structure.

**Figure 3 nanomaterials-11-02327-f003:**
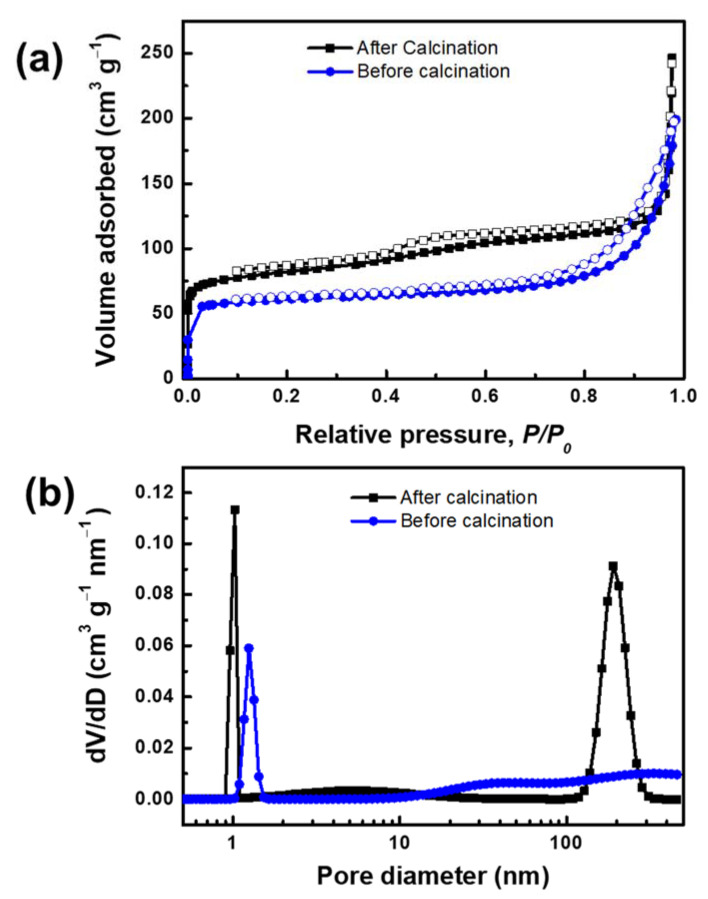
(**a**) N_2_ adsorption and desorption isotherm and (**b**) pore size distribution of MLSil-1Ns-15d confirming the microporous and mesoporous structure of the sample before and after calcination at 550 °C for 5 h.

**Figure 4 nanomaterials-11-02327-f004:**
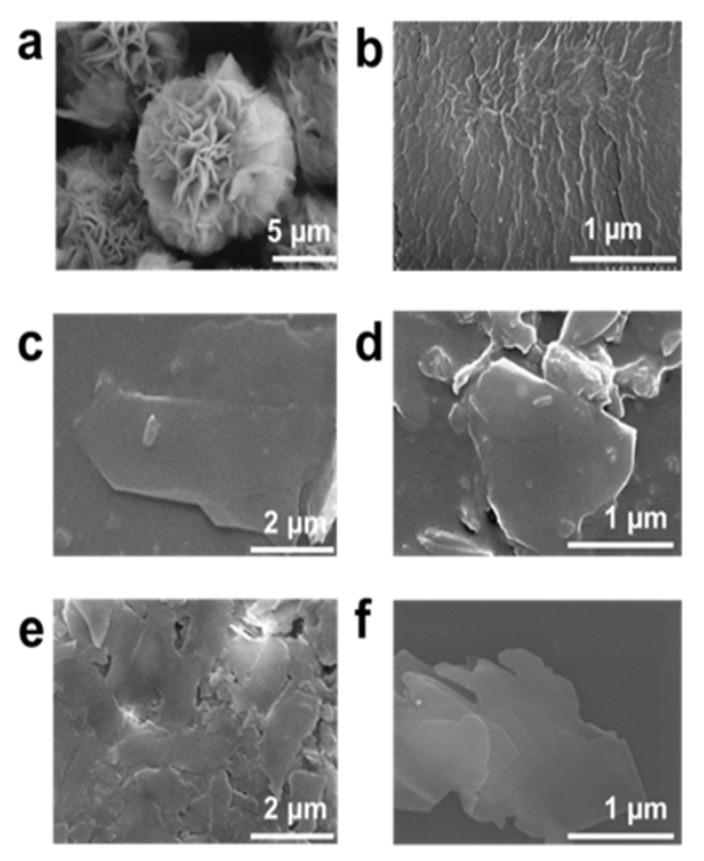
Effect of planetary milling and purification of exfoliated MFI zeolite nanosheets, (**a**) before the milling of nanosheets, (**b**) after planetary milling for 15 min the presence of PVP surfactant, (**c**) after the centrifugation for one time. The surfactant embedded in the MLSil-1Ns was still present in the solution. The large part of interlocked nanosheets exfoliated in few layers, (**d**) major part of surfactant has been removed from the exfoliated nanosheets suspension however some part was found remaining even after two times centrifugation, (**e**,**f**) cleaned and purified exfoliated nanosheets were obtained after samples were centrifuged three times.

**Figure 5 nanomaterials-11-02327-f005:**
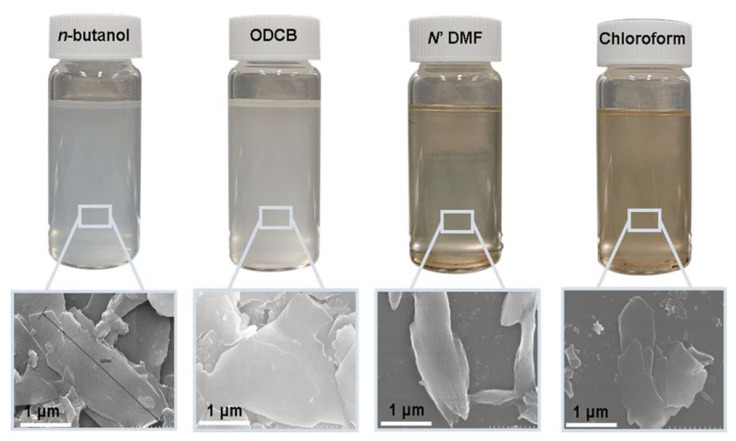
Suspension of exfoliated nanosheets in organic solvents in order of most stable suspension to agglomerated and stacked nanosheets (*n*-butanol > *o*-DCB > *N*′*N*′DMF > chloroform).

**Figure 6 nanomaterials-11-02327-f006:**
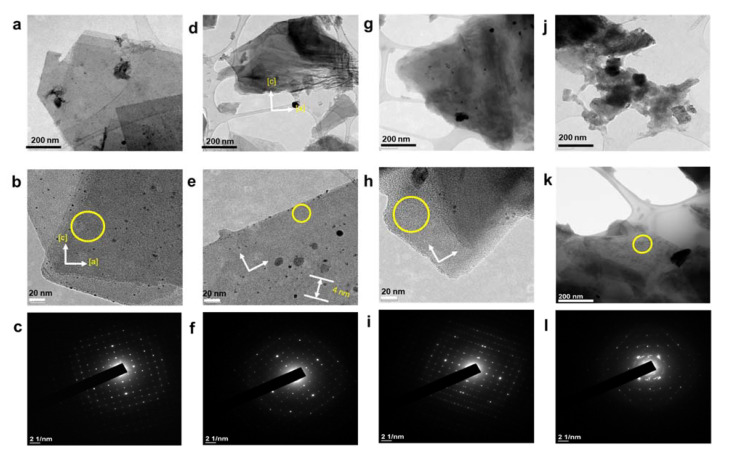
TEM images of exfoliated nanosheets after purification by repeatedly washing with toluene under centrifugation, (**a**–**c**) 10 wt.% suspension of nanosheets dispersed in *n*-butanol and corresponding SAED patterns of suspended nanosheets shows the few or single layers of nanosheets. The nanosheets were freely suspended in *n*-butanol solvent. (**d**–**f**), 10 wt.% suspension of nanosheets dispersed in *o*-DCB and the corresponding SAED pattern shows the few or single layers of nanosheets, (**g**–**i**) 10 wt.% suspension of nanosheets dispersed in *N*′*N*′DMF and the corresponding SAED patterns, (**j**–**l**) 10 wt.% suspension of nanosheets dispersed in chloroform and the corresponding SAED patterns.

**Figure 7 nanomaterials-11-02327-f007:**
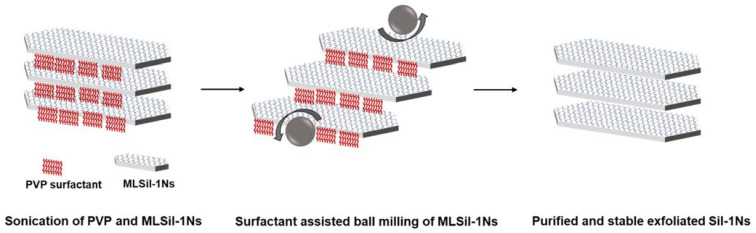
Schematic diagram for the synthesis of MLSil-1Ns, exfoliation and purification of nanosheets.

**Table 1 nanomaterials-11-02327-t001:** Molecular weight, kinetic diameter, steric hindrance and molecular structure of organic solvents.

Solvents	Molecular Weight	Kinetic Diameter	Steric Hindrance (A Values)	Molecular Structure
	g mol^−1^	Å	kcal mol^−1^	
*o*-DCB	147.01	5.63	3.0	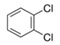
*n*-butanol	74.12	5.14	1.70	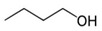
*N*′*N*′DMF	73.09	4.90	0.67	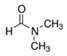
Chloroform	119.30	4.72	0.48	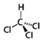

**Table 2 nanomaterials-11-02327-t002:** List of reported exfoliated zeolite nanosheets methods.

Layered Materials	Exfoliation Method	Dispersant/Surfactant	Ref.
ITQ-1	Swelling	^1^ CTAB	[15]
MFI nanosheets	Melt compounding	Polystyrene	[15]
MCM-22	Swelling	CTAB	[17]
MCM-22	Milling and swelling	^2^ HTPB	[18]
MFI nanosheets	Milling and sonication	HTPB	[18]
MLSil-1Ns	Sonication and milling	^3^ PVP	This study

^1^ CTAB: Cetyl trimelthyl ammonium bromide, ^2^ HTPB: hydroxyl-terminated polybutadienes, ^3^ PVP: polyvinylpyrrolidone.

## Data Availability

Data can be available upon request from the authors.

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
