# Peer review of "Two-Dimensional MFI Zeolite Nanosheets Exfoliated by Surfactant Assisted Solution Process"

_nanomaterials, 2021, doi:10.3390/nano11092327_

Round 1

Reviewer 1 Report

The manuscript focuses on characterization and fabrication of 2D MFI zeolite nanosheets using PVP as an exfoliation agent and sonification/mechanochemical treatment. The research is original and could be interesting for the readership of the Nanomaterials. The manuscript is well-written, the language is concise and understandable.

There are some concerns that need to be addressed:

1) Please indicate the purity and suppliers for all compounds used in this study. Was the purification and drying of solvents and reagents carried out or they were used as received?

2) Some important details on ball-milling are missing: what was the ball-to-powder ratio (BPR) and did the synthesis program include cooling breaks between the milling?

3) The information about MLsil-1Ns-15d calcination (550°C for 5 h) is found only in footnote to the Figure 3, please add this information in the main text as well (e.g. in Experimental part).

Author Response

Thank you for your comments. I tried to cover all your comments. Your comments were helpful to improve my manuscript. 

Reviewer 2 Report

In this work, the authors used polyvinlypyrrolidone as an exfoliation agent to synthesize two-dimensional zeolite nanosheets. The effects of structure directing agent, exfoliation agent, and organic solvents on the stability of exfoliated nanosheets were investigated. This work demonstrates that PVP-assisted liquid exfoliation is a promising approach to prepare the 2D zeolite nanosheet. In general, this work is of interest to the audience of the Nanomaterials. I have few comments which need to be addressed before acceptance.

Page 2: change multilamellar MFI nanosheets (MLSil- 1Ns) to multilamellar silicalite-1 nanosheets (MLSil-1Ns).

Page 3: “The bi-functional organic surfactant (di-quaternary ammonium type, C22-6-6-Br) was synthesized by the method developed by Ryoo and co-workers.” For clarity purpose, the complete chemical structure of C22-6-6 should be specified, i.e., C22 H45 -N+ (CH3 )2 -C6 H12 -N+ (CH3 )2 -C6 H13 (denoted as C22-6-6 ).

The denotation of SDA and OSDA is not defined in the main text, though they were defined in the Supplementary Information.

Change MLsil-1Ns to MLSil-1Ns and keep these denotations consistent.

Page 4: Figure 1: what does XRD peak {1000} mean?

Page 6: Remove the black solid box surrounding the labels a and b.

Page 8: N’, N’ dimethyl formamide (N’ DMF). Should be N’N’DMF.

The denotation of SAED is not defined.

Page 10: “The repulsion force, in other words the stable dispersion can be achieved by in- creasing the charge on nanoparticle or by adding larger molecular side chains to induce steric hindrance.”

The repulsion force is not dispersion.

Please check the grammar and rephrase the following sentences.

“In the case of o-DCB, the steric hindrance generates from strong electron cloud from the benzene ring and has higher steric value of 3.0 kCal mol-1 (Table 1).”

“The hindrance come from extended alkyl-chain results in strong barrier and avoid the nanosheets agglomeration.”

“chloroform and N’DMF exhibit smaller molecular as well as lower steric hindrance.” Change molecular to molecular weight. Change N’DMF to N’ N’DMF in Figure 5.

Figure 6 caption: Please rephrase the sentence “shows the polycrystalline patterns due to the re-stacking and agglomeration of because of chloroform effect.”

Figure 6k: The two scale bars overlapped with each other.

“Therefore, the nanosheets agglomerates and settled down in these solvents.” Please check the grammar.

There should be a citation for Table 1.

Page 12: Figure 7 needs to be revised.

Figure 7a needs a better representation.  

Why PVP is shown in different colors in Figure 7c?

Figure 7f and 7h are missing according to the figure caption.

Author Response

(The authors gave the same response as above.)

Reviewer 3 Report

Cho et. al, have provided  a PVP-assisted liquid exfoliation strategy for the preparation of 2D zeolite nanosheets, however, this methods is a conventional technology for exfoliation of 2D nanosheets. The highlights of this work is the SAED patterns  of the 2D zeolite nanosheets in different solvent systems, the authors are expected to get deeper investigation into the relationship between the variation of SAED patterns and  solvent systems.

Author Response

(The authors gave the same response as above.)

Round 2

Reviewer 3 Report

The revised manuscript is acceptable.